# Synthesis, Characterization, and the Antioxidant Activity of Phenolic Acid Chitooligosaccharide Derivatives

**DOI:** 10.3390/md20080489

**Published:** 2022-07-28

**Authors:** Yan Sun, Xia Ji, Jingmin Cui, Yingqi Mi, Jingjing Zhang, Zhanyong Guo

**Affiliations:** 1Key Laboratory of Coastal Biology and Bioresource Utilization, Yantai Institute of Coastal Zone Research, Chinese Academy of Sciences, Yantai 264003, China; sunyan20@mails.ucas.ac.cn (Y.S.); jmcui@yic.ac.cn (J.C.); yqmi@yic.ac.cn (Y.M.); jingjingzhang@yic.ac.cn (J.Z.); 2University of Chinese Academy of Sciences, Beijing 100049, China; 3School of Pharmacy, Qilu Medical University, Zibo 255300, China

**Keywords:** chitooligosaccharide derivatives, phenolic acid, antioxidant activity

## Abstract

A series of phenolic acid chitooligosaccharide (COS) derivatives synthesized by two mild and green methods were illuminated in this paper. Seven phenolic acids were selected to combine two kinds of COS derivatives: the phenolic acid chitooligosaccharide salt derivatives and the phenolic-acid-acylated chitooligosaccharide derivatives. The structures of the derivatives were characterized by FT-IR and ^1^H NMR spectra. The antioxidant experiment results in vitro (including DPPH-radical scavenging activity, superoxide-radical scavenging activity, hydroxyl-radical scavenging ability, and reducing power) demonstrated that the derivatives exhibited significantly enhanced antioxidant activity compared to COS. Moreover, the study showed that the phenolic acid chitooligosaccharide salts had stronger antioxidant activity than phenolic-acid-acylated chitooligosaccharide. The cytotoxicity assay of L929 cells in vitro indicated that the derivatives had low cytotoxicity and good biocompatibility. In conclusion, this study provides a possible synthetic method for developing novel and nontoxic antioxidant agents which can be used in the food and cosmetics industry.

## 1. Introduction

Having unpaired electronics, free radicals are unstable, short lived, and highly reactive atoms or molecules. They can be classified into reactive oxygen species (ROS) and reactive nitrogen species (RNS). ROS are the most common radicals in living organisms, including superoxide (O_2_^•−^), the oxygen radical (O_2_^••^), the hydroxyl radical (OH^•^), the alkoxy radical (RO^•^), the peroxyl radical (ROO^•^), etc. Free radical reactions play an important role in life science. In general, the production and elimination of free radicals in human body are in a state of dynamic equilibrium, which is the key to keeping healthy. However, with aging and the formation of some unhealthy habits, the free radicals may exceed normal levels and cause oxidative damage. This is harmful to the body, because free radicals can attack the biomolecules, including lipids, proteins, and DNA; accelerate aging; generate age-related diseases; cause irreversible damage; and even lead to cancer and neurodegeneration diseases [1,2,3,4]. In addition, free radicals in food can lead to lipid oxidation, resulting in a decrease in food quality, a decrease in nutritional value and color changes. Therefore, it is necessary to synthesize and develop novel and green antioxidants which can be used in these fields, and researchers are paying this more and more attention.

Being widespread in nature, phenolic acids are aromatic secondary metabolites and one of the most significant bioactive substances that are present in abundant plant sources. Phenolic acids can be divided into hydroxybenzoic acids and hydroxycinnamic acids according to the differences in carbon frameworks and the positions and numbers of hydroxyl groups on the aromatic ring. The most common hydroxybenzoic acids are: gallic acid, *p*-hydroxybenzoic acid, salicylic acid, ellagic acid, gentisic acid, protocatechuic acid, syringic acid, and vanillic acid. Meanwhile, the most common hydroxycinnamic acid and its derivatives are: *p*-coumaric acid, cinnamic acid, caffeic acid, ferulic acid, sinapic acids, ferulic acid, and *p*-hydroxycinnamic acid [5]. Phenolic acids are phenotypic natural antioxidants, which can exert antioxidant effects by delivering hydrogen to free radicals and interfering with the chain-propagation reactions [6]. The antioxidant capacity of phenolic acids is also linked to the number and arrangement of hydroxyl groups [7]. Previous study have shown that phenolic acids also have other biological activities, such as anti-tumor activity, antibacterial activity, and antiviral activity [5,8]. Based on these beneficial biological activities, phenolic acids have become potential novel materials that can be used in the fields of food, medicine, and cosmetics. However, the poor water solubility of phenolic acids is an impediment to further applications.

Chitin, mainly from marine resources, is the second most abundant natural biopolymer on earth after cellulose [9]. Chitosan, as the deacetylated product of chitin, composed of beta1-4 linked *D*-glucosamine (deacetylated unit) and *N*-acetyl-*D*-glucosamine (acetylated unit), is the only cationic polymer in nature [10]. Studies have shown that chitosan has many beneficial biological activities, such as antioxidant activity, antibacterial activity, biodegradability, and biocompatibility, but the poor solubility of chitosan greatly restricts its further application. Chitooligosaccharide (COS), as the degraded product of chitosan (the degree of polymerization varies from 2 to 20 [11]), has good water solubility and enhanced biological activity. Compared to chitosan, COS not only has better physicochemical properties, such as excellent water solubility, biodegradability, and biocompatibility, but also has various biological activities, including anti-inflammation, anti-bacteria, immunomodulation, neuroprotection, and so on [12,13]. In general, COS is prepared by chemical or enzymatic hydrolysis methods [14] by dissolving chitosan in various acids. Additionally, COS prepared in this way often exits in the form of COS acid salts. At present, the common COS salts include chitooligosaccharide acetate, chitooligosaccharide hydrochloride, and chitooligosaccharide lactate. According to studies, different COS salt forms may have different activities [15,16]. However, chitooligosaccharide phenolic acid salts has not been reported. In addition, COS has amino and hydroxy groups, which can be functionalized with many substances to further improve the bioactivity [11,17]. Therefore, the research on COS is attracting more and more attention and interest. Chemical modification of COS to obtain more biologically active substances is becoming the research emphasis.

There have been many studies on the preparation of chitooligosaccharide derivatives grafted with phenolic acids. Derivatives were formed by various methods and had various biological activities. Tae et al. selected eight kinds of phenolic acids (*p*-hydroxybenzoic acid, p-coumaric acid, protocatechuic acid, caffeic acid, vanillic acid, ferulic acid, syringic acid, sinapinic acid) to prepare eight acylated COS derivatives via a DCC catalytic system. The antioxidant activity of the derivatives was stronger than that of COS, indicating that the modification of phenolic acids could enhance the antioxidant activity of COS [18,19]. Thanh et al. prepared gallate-chitooligosaccharide derivatives with significantly improved antioxidant and anti-inflammatory activity using the DCC catalytic system [20]. Moreover, a conjugated method of synthesizing COS with phenolic acids mediated by the VC/H_2_O_2_ catalytic system was reported. COS firstly reacted with VC and H_2_O_2_ to generate free radicals, and then combined with five phenolic acids (gallic, caffeic, ferulic acids, epigallocatechin gallate, and catechin) to form five derivatives. The antioxidant, antidiabetic, and antimicrobial activities of these derivatives were measured, and the results showed that the catechin-chitooligosaccharide conjugated compound had the highest activities [21]. In another study, three phenolic acids (caffeic acid, ferulic acid, sinapic acid) were chosen to form three chitosan acylated derivatives by the VC/H_2_O_2_ catalytic system, and their antioxidant and antimicrobial activities were tested in vitro. The results demonstrated that the antioxidant and antimicrobial activities of derivatives were stronger than chitosan [22]. Li et al. prepared four phenolic-functionalized chitosan derivatives by hydroxybenzaldehyde (4-hydroxybenzaldehyde, 3-hydroxybenzaldehyde, 3,4-dihydroxy-benzaldehyde, and 2,3-dihydroxy-benzaldehyde) reacted with 6-N-(aminoethyl)-N-trimethyl quaternary ammonium chitosan, and the antioxidant activity of these derivatives was obviously enhanced compared to chitosan [23]. In another study, Li et al. used three hydroxycinnamic acids (*p*-coumaric acid, ferulic acid, and sinapic acid) to modify chitosan quaternary ammonium and obtained chitosan derivatives which had higher antioxidant and antitumor activities [24]. As mentioned above, many studies have focused on the modification of chitosan and COS by phenolic acids to obtain conjugated compounds with increased activity. However, there are more studies on acylated conjugated compounds and fewer on phenolic acids’ salt derivatives. In addition, the EDC/NHS catalytic system, which can catalyze acylation reactions, is seldom reported. Therefore, in this study, seven kinds of phenolic acid (gallic acid monohydrate, ferulic acid, *p*-coumaric acid, caffeic acid, protocatechuic acid, sinapic acid, and salicylic acid) and chitooligosaccharide lactate were selected to prepare phenolic acid chitooligosaccharide derivatives in two ways. Additionally, the antioxidant activities of two series of derivatives were tested and compared. By the first synthetic route, phenolic acid chitooligosaccharide salts were produced, and COS was modified with phenolic acid by ionic bonding via a mild and non-toxic route (ion exchange). The protonated amino group at position 2 of COS bound with carboxyl negative ions on phenolic acids to form ionic bonds. Seven derivatives were successfully synthesized: gallic acid monohydrate-chitooligosaccharide conjugates (GLCOS), ferulic acid-chitooligosaccharide conjugates (FUCOS), *p*-coumaric acid-chitooligosaccharide conjugates (CMCOS), caffeic acid-chitooligosaccharide conjugates (CFCOS), protocatechuic acid-chitooligosaccharide conjugates (PCCOS), sinapic acid-chitooligosaccharide conjugates (SPCOS), and salicylic acid-chitooligosaccharide conjugates (SYCOS). As the second synthetic route, we produced phenolic-acid-acylated chitooligosaccharide derivatives, in which COS was modified with phenolic acid by amido bonding via the EDC/NHS catalytic system [25]. Seven derivatives was successfully synthesized: gallic-acid-monohydrate-acylated chitooligosaccharide conjugates (GL-COS), ferulic-acid-acylated chitooligosaccharide conjugates (FU-COS), *p*-coumaric-acid-acylated chitooligosaccharide conjugates (CM-COS), caffeic-acid-acylated chitooligosaccharide conjugates (CF-COS), protocatechuic-acid-acylated chitooligosaccharide conjugates (PC-COS), sinapic-acid-acylated chitooligosaccharide conjugates (SP-COS), and salicylic-acid-acylated chitooligosaccharide conjugates (SY-COS). The structures of these derivatives were characterized by FT-IR and ^1^H NMR. The antioxidant activity of COS derivatives, including the DPPH radical scavenging activity, superoxide-radical scavenging activity, hydroxyl radical scavenging ability, and reducing power were evaluated and compared in vitro. What is more, the cytotoxicity assay of L929 cells was implemented to explore the biocompatibility of products. This study may provide novel insights into developing the usage fields of COS.

## 2. Materials and Methods

### 2.1. Materials

Chitooligosaccharide (MW 2000 Da, the degree of deacetylation was 95%) was purchased from Shandong Weikang Biomedical Technology Co., Ltd. (Linyi, China). L929 cells (GDC0034) were obtained from the China Center for Type Culture Collection. Gallic acid monohydrate, ferulic acid, *p*-coumaric acid, caffeic acid, protocatechuic acid, sinapic acid, salicylic acid 1-(3-dimethylaminopropyl)-3-ethylcarbodiimide hydrochloride (EDC•HCl), and *N*-hydroxysuccinimide (NHS), and other chemical reagents, were purchased from Sigma-Aldrich Chemical Corp. (Shanghai, China). All chemical reagents were AR, obtained from commercial sources, and used as received.

### 2.2. Synthesis of Chitooligosaccharide Derivatives

#### 2.2.1. Synthesis of Phenolic Acid Chitooligosaccharide Salts

First, 30 mmol amounts of phenolic acid (gallic acid monohydrate, ferulic acid, *p*-coumaric acid, caffeic acid, protocatechuic acid, sinapic acid, salicylic acid) were dissolved into deionized water and anhydrous ethanol in a flask at 65 °C. The amount of solvent varied from 30 to 50 mL due to the different solubility of phenolic acids. Then, 10 mmol of COS was dissolved in 10 mL deionized water and added to phenolic acid solution drop by drop. After that, the reaction system was stirred at 65 °C for 24 h. Then, the anhydrous ethanol was used to precipitate and wash the products. The phenolic acid COS salts were obtained after freeze drying at −50 °C for 24 h.

#### 2.2.2. Synthesis of Phenolic Acid Acylated Chitooligosaccharide

First, 25 mmol of phenolic acid (gallic acid monohydrate, ferulic acid, *p*-coumaric acid, caffeic acid, protocatechuic acid, sinapic acid, salicylic acid) was dissolved into deionized water or anhydrous ethanol in a flask at room temperature. The amount of solvent varied from 30 to 50 mL because of the solubility difference. Additionally, 5% NaOH was used to adjust the pH of solution to 5. Then, 50 mmol of EDC and NHS was added into the solution with stirring for 2 h. After that, 10 mmol COS was added to it, and it was stirred for 24 h at room temperature. Finally, the anhydrous ethanol was used to precipitate and wash the products. The phenolic-acid-acylated COS was obtained by freeze drying at −50 °C for 24 h [26].

### 2.3. Analytical Methods

#### 2.3.1. Fourier Transform Infrared (FT-IR) Spectroscopy

In order to characterize the changes in the products, the FT-IR spectra of products were recorded by a Nicolet IS50 Fourier Transform Infrared Spectrometer (Thermo, Waltham, MA, USA), provided by Thermo Fisher Scientific (Shanghai, China). The parameter settings were a resolution of 4.0 cm^−1^ and a 400–4000 cm^−1^ range. The measured samples were mixed with KBr disks and scanned 16 times at 25 °C for measurement.

#### 2.3.2. Nuclear Magnetic Resonance (NMR) Spectroscopy

The ^1^H NMR spectra of products were recorded by an AVIII-500 Spectrometer (500 MHz, Bruker Switzerland) at 25 °C. D_2_O was used as the solvent for all tested samples. Chemical shift values are given in δ (ppm).

### 2.4. Antioxidant Activity Assay

#### 2.4.1. DPPH-Radical Scavenging Activity Assay

1,1-Diphenyl-2-picryl-hydrazyl (DPPH), as a stable free radical, is one of the most commonly used free radicals for antioxidant experiments in vitro [27]. In accordance with Sun and Bajpai [28,29], the DPPH-radical scavenging activity of products was tested as follows. Firstly, all the tested samples with an initial concentration of 10 mg/mL (0.03, 0.06, 0.12, 0.24, 0.48 mL) and a 2.00 mL ethanol solution of DPPH (180 µM) were incubated for 2 min. At the same time, DPPH was replaced by 2.00 mL anhydrous ethanol in the control group, and the sample was replaced by 1.00 mL deionized water in the blank group. Then, the absorbance of solutions was measured at 517 nm. Three replicates for every sample concentration were measured, and the scavenging effect was calculated by the following equation:$$\mathrm{Scavenging}\;\mathrm{effect}\;\left(\%\right)=\left[1-\frac{A_{sample\;517\;\mathrm{nm}}-A_{sample\;517\;\mathrm{nm}}}{A_{blank\;517\;\mathrm{nm}}}\right]\times 100$$
where *A_sample_*
_517 nm_, *A_control_*
_517 nm_, and *A_blank_*
_517 nm_ represent, respectively, the absorbance of the sample, the control, and the blank.

#### 2.4.2. Superoxide-Radical Scavenging Activity Assay

According to Yattra [30] and Zhang [31], the method of superoxide-radical scavenging activity measurement was modified slightly and implemented as follows: 5.16 × 10^−4^ mol/L of reduced nicotinamide adenine dinucleotide (NADH) solution, 3.37 × 10^−5^ mol/L of nitrotetrazolium blue chloride (NBT) solution, and 5.87 × 10^−5^ mol/L phenazine methosulfate (PMS) solution in Tris-HCl buffer (pH = 8.0) were prepared for the experiment. The reaction system including the tested samples with initial concentration of 10 mg/mL (0.03, 0.06, 0.12, 0.24, 0.48 mL), NADH (0.50 mL), NBT (0.50 mL), and PMS (0.50 mL), was incubated for 5 min at 25 °C. Meanwhile, in the control group, NADH was replaced by equivalent Tris-HCl buffer, and the sample was replaced by deionized water in the blank group. The absorbance was measured at 560 nm. Three replicates for every sample concentration were measured, and the scavenging effect was calculated by the following equation:$$\mathrm{Scavenging}\;\mathrm{effect}\;\left(\%\right)=\left[1-\frac{A_{sample\;560\;\mathrm{nm}}-A_{control\;560\;\mathrm{nm}}}{A_{blank\;560\;\mathrm{nm}}}\right]\times 100$$
where A*_sample_*
_560 nm_, A*_control_*
_560 nm_, and A*_blank_*
_560 nm_ represent, respectively, the absorbance of the sample, the control, and the blank.

#### 2.4.3. Hydroxyl-Radical Scavenging Activity Assay

The hydroxyl-radical scavenging activity of products was measured as follows. The reaction system, including the tested samples with initial concentration of 10 mg/mL (0.045, 0.09, 0.18, 0.36, 0.72 mL), EDTA-Fe^2+^ (220 µM), safranine T (0.23 µM), and H_2_O_2_ (60 µM) in potassium phosphate buffer (pH = 7.4), was incubated at 37 °C for 30 min. Moreover, in the control group, the sample was replaced by deionized water and the H_2_O_2_ was replaced by equivalent potassium phosphate buffer. The sample was replaced by deionized water in the blank. The absorbance of the reaction system was measured at 520 nm. Three replicates for every sample concentration were measured, and the scavenging effect was calculated by the following equation:$$\mathrm{Scavenging}\;\mathrm{effect}\;\left(\%\right)=\left[\frac{A_{sample\;520\;\mathrm{nm}}-A_{blank\;520\;\mathrm{nm}}}{A_{control\;520\;\mathrm{nm}}-A_{blank\;520\;\mathrm{nm}}}\right]\times 100$$
where A*_sample_*
_520 nm_, A*_control_*
_520 nm_, and A*_blank_*
_520 nm_ represent, respectively, the absorbance of the sample, the control, and the blank [32].

#### 2.4.4. Reducing Power Assay

The reducing power of products was tested as follows. Firstly, all the tested samples with initial concentration of 10 mg/mL (0.03, 0.06, 0.12, 0.24, 0.48 mL) and 0.50 mL potassium ferricyanide (1%, *w*/*v*) dissolved in sodium phosphate buffer (0.2 M, pH = 6.6) were incubated at 50 °C for 20 min. Then, 1.00 mL trichloroacetic acid (10%, *w*/*v*) was added to terminate the reaction. After centrifugation, 0.75 mL supernatant, 0.60 mL deionized water, and 0.15 mL ferric chloride solution (0.1%, *w*/*v*) were incubated for 10 min at room temperature. The absorbance of the reaction system was measured at 700 nm. Three replicates for every sample concentration were measured [33].

### 2.5. Cytotoxicity Assay

L929 cells were used in vitro to test the cytotoxicity of derivatives by Cell Counting Kit-8 (CCK-8). It is a rapid and highly sensitive test kit based on WST-8 and widely used in the detection of cell cytotoxicity. According to Cai [34] and Zhang [35], the method of cytotoxicity analysis was modified slightly and implemented as follows. The resuscitated cells were cultured in RPMI medium (including 1% streptomycin, 1% penicillin, 10% fetal bovine serum) at 37 °C under a 5% CO_2_ atmosphere. After two passages, the new resuscitated cells were transferred to 96-well plates and incubated in a CO_2_ incubator for 24 h. Then, the products with different concentrations (1000 μg/mL, 500 μg/mL, 250 μg/mL, 125 μg/mL, and 62.5 μg/mL) were added into it and cultured for 24 h. After that, 100 μL CCK-8 reagent was added to each well, and the absorbance was detected at 450 nm with a microplate reader after 4 h. Meanwhile, the blank group only included the RPMI medium and CCK-8 solution, and the control group contained CCK-8 solution and cells. The experiment was repeated 3 times in parallel for each concentration. The cell viability was calculated by the following equation:$$\mathrm{Cell}\;\mathrm{viability}\;\left(\%\right)=\left[\frac{A_{sample\;450\;\mathrm{nm}}-A_{blank\;450\;\mathrm{nm}}}{A_{control\;450\;\mathrm{nm}}-A_{blank\;450\;\mathrm{nm}}}\right]\times 100$$
where A*_sample_* _450 nm_, A*_blank_* _450 nm_, and A*_control_* _450 nm_ represent the absorbance of the samples, the blank, and the control at 450 nm, respectively.

### 2.6. Statistical Analysis

All the experiments were performed in triplicate, and the data are presented as means ± standard deviations (SDs). The significant difference was determined by Scheffe’s multiple range test. The results with a level of *p* < 0.05 were considered statistically significant.

## 3. Results and Discussion

### 3.1. Chemical Synthesis and Characterization

The synthetic strategy of phenolic acid chitooligosaccharide salt derivatives is shown in Route 1 of Figure 1. Based on the electrostatic attraction of positive and negative charges, the amino group of COS protonated under acidic conditions and bound with carboxyl negative ions on phenolic acids to form ionic bonds. The structures of all samples were characterized by FT-IR and ^1^H NMR.

The synthetic procedure of phenolic-acid-acylated chitooligosaccharide derivatives is shown in Route 2. EDC can react with carboxyl groups processed in phenolic acids and form amine-reactive O-acylisourea. With the presence of NHS, the amine-reactive O-acylisourea can immediately form an amine-reactive NHS ester, which is a relatively stable intermediate. After COS was added into the reaction system, the amine-reactive NHS ester could react with the amino groups on COS and form an amide bond [25]. Based on the above principle, the phenolic acids were firstly activated by the EDC/NHS reaction system and then combined with COS to form acylated derivatives. The structure of each sample was verified by FT-IR and ^1^H NMR spectroscopy analysis.

#### 3.1.1. FT-IR Spectra

The infrared spectra of COS and phenolic acid chitooligosaccharide salts are shown in Figure 1. For COS, according to Mourya [12], as exhibited in the figure, the band at 3300–3000 cm^−1^ is concluded to be O-H and N-H bending, and the band of 2924–2840 cm^−1^ is the stretching vibration of O-H. In addition, the characteristic spike at 1582 cm^−1^ is assigned to an amino group. The absorption peak at 1415 cm^−1^ is the deformation vibrations of -CH_2_ and -CH_3_. The absorption peak at 1377 cm^−1^ can attributed to an amide III bond. Additionally, the absorption peaks at 1084 and 1040 cm^−1^ are C-O-C stretching vibrations. After reacting with various phenolic acids, the spectra of the derivatives changed obviously. The FT-IR peaks of GLCOS (1455, 1537 cm^−1^), FUCOS (1516 cm^−1^), CMCOS (1520, 1541 cm^−1^), CFCOS (1537, 1555 cm^−1^), PCCOS (1468, 1555 cm^−1^), SPCOS (1520, 1541, 1555 cm^−1^), and SYCOS (1456, 1486 cm^−1^) can be ascribed to the characteristic absorption of C-C stretching vibrations of the benzene rings. Meanwhile, the peaks appearing at GLCOS (795, 856 cm^−1^), FUCOS (841 cm^−1^), CMCOS (852, 903 cm^−1^), CFCOS (770, 815, 854 cm^−1^), PCCOS (719, 776, 901 cm^−1^), SPCOS (774, 897 cm^−1^), and SYCOS (767, 860, 895 cm^−1^) can be contributed to the C-H out of plane bending vibration of the benzene rings [36]. The derivatives kept the characteristic peaks of COS (3300, 2840, 1590, 1080, 1040 cm^−1^). In summary, the new peaks can preliminarily prove the successful synthesis of phenolic acid COS salt derivatives.

The infrared spectra of COS and phenolic-acid-acylated chitooligosaccharide derivatives are shown in Figure 2. Compared to COS, the reaction of COS with phenolic acids led to an evident new peak at 1650 cm^−1^, which could be attributed to the C=O stretch of the amide bond. Furthermore, the absorption spikes at 1565, 1539, 1553, 1519, 1536, 1531, 1533, 1555, and 1535 cm^−1^ in the infrared spectra of GL-COS, FU-COS, CM-COS, CF-COS, PC-COS, SP-COS, and SY-COS, respectively, can be assigned to the characteristic absorption of C-C stretching vibration of the benzene rings. The spikes appearing at 700−900 cm^−1^ of GLCOS, FUCOS, CFCOS, PCCOS, SPCOS, and SYCOS can be attributed to the C-H out of plane bending vibration of the benzene rings. To sum up, all the evidence indicates the successful synthesis of the products.

#### 3.1.2. ^1^H NMR Spectra

The ^1^H NMR spectra of phenolic acid chitooligosaccharide salt derivatives are shown in Figure 3, and those of the phenolic-acid-acylated chitooligosaccharide derivatives are shown in Figure 4. The chemical shift at δ 4.79 ppm represents the D_2_O solvent. For COS, the chemical shifts can be attributed as follows: (H1): δ 4.48 ppm; (H3)–(H6): δ 3.21–4.08 ppm; (H2): δ 2.69 ppm. For GLCOS and GL-COS, the new peaks appear at δ 6.87 ppm (a). For FUCOS, the new chemical shifts are present at δ 7.17 ppm (a), δ 7.11 ppm (b), δ 6.97 ppm (c), δ 6.77 ppm (d), δ 6.24 ppm (e), and δ 4.12 ppm (f), along with the same ones as FU-COS. For CMCOS and CM-COS, the new characteristic peaks are δ 6.23 ppm (a), δ 7.19 ppm (b), δ 7.38 ppm (c, d), and δ 6.78 (e, f) [37]. For CFCOS and CF-COS, the new chemical shifts are at δ 6.18 ppm (a), δ 7.13 ppm (b), δ 7.00 ppm (c), δ 6.91 ppm (d), and δ 6.77 ppm (e). For PCCOS, the new characteristic peaks are δ 7.23 ppm (a, b), 6.77 ppm (c), and the same ones as PC-COS. For SPCOS and SP-COS, the new chemical shifts appear at δ 6.41 ppm (a), δ 7.14 ppm (b), δ 6.25 ppm (c, d), and δ 4.11 ppm (e). For SYCOS, the new chemical shifts appear at δ 7.66 ppm (a), δ 6.79 ppm (b, d), δ 7.30 ppm (c), and the same ones as SY-COS. Through the above analysis, the FT-IR data and the ^1^H NMR data demonstrate the successful synthesis of the products.

### 3.2. Antioxidant Activity

After characterizing the structures of two series of COS phenolic acid derivatives, their antioxidant activity was also tested. Several in vitro experiments, including the DPPH-radical scavenging experiment, the superoxide-radical scavenging experiment, the hydroxyl-radical scavenging experiment, and the reducing power experiment were carried out. Ascorbic acid (VC) was chosen as the positive control in all experiments. The experimental results are shown in Figure 5, Figure 6, Figure 7 and Figure 8.

#### 3.2.1. Scavenging Ability against DPPH Radicals

The DPPH-radical scavenging ability of COS and phenolic acid chitooligosaccharide salt derivatives is shown in Figure 5a. Several conclusions can be drawn from the figure. Firstly, as we can see, all the derivatives have better scavenging ability in the tested concentration range compared with COS. Secondly, the scavenging activity of all derivatives increased in a dose-dependent manner. Thirdly, among these derivatives, GLCOS, CFCOS, and PCCOS have remarkable scavenging ability—close to 100% at the tested concentrations. Moreover, at the concentration of 1.60 mg/mL, the scavenging indices of samples were as follows: VC—100%, GLCOS—96.15%, FUCOS—96.50%, CMCOS—100%, CFCOS—96.03%, PCCOS—100%, SPCOS—96.01%, SYCOS—100%, COS—34.52%.

The DPPH-radical scavenging ability of COS and phenolic-acid-acylated chitooligosaccharide derivatives is shown in Figure 5b. As we can see in the figure, all derivatives have stronger scavenging ability in the tested concentrations than COS. Additionally, the scavenging effects of all derivatives were more than 90% at the concentration of 1.60 mg/mL. In summary, the incorporation of phenolic acid could improve the DPPH scavenging activity of COS significantly.

#### 3.2.2. Scavenging Ability against Superoxide Radicals

The superoxide-radical scavenging ability of COS and phenolic acid chitooligosaccharide salt derivatives is shown in Figure 6a. Similarly to the DPPH scavenging ability, all the derivatives have stronger scavenging ability than COS at the tested concentrations. For example, at the concentration of 0.80 mg/mL, the scavenging indices of the samples were listed as follows: VC—100%, GLCOS—100%, FUCOS—72.97%, CMCOS—80.36%, CFCOS—89.73%, PCCOS—89.96%, SPCOS—76.47%, SYCOS—69.64%, COS—35.62%. The data indicate that modification of COS with phenolic acids could improve its scavenging activity obviously.

Figure 6b shows the superoxide-radical scavenging ability of COS and phenolic-acid-acylated chitooligosaccharide derivatives. The scavenging activity of samples became stronger with increasing concentration. For instance, when the concentration of GL-COS was 0.10, 0.20, 0.40, 0.80, and 1.60 mg/mL, the scavenging effects were 44.82%, 64.77%, 84.82%, 94.47%, and 100% respectively. The scavenging indices of phenolic-acid-acylated chitooligosaccharide derivatives were lower than those of the phenolic acid chitooligosaccharide salt derivatives, presumably because the phenolic acid chitooligosaccharide salt derivatives have more dense positive charges.

#### 3.2.3. Scavenging Ability against Hydroxyl Radicals

Figure 7a shows the hydroxyl-radical scavenging ability of COS and phenolic acid chitooligosaccharide salt derivatives. It can be observed that with the increase in concentration, the scavenging effects of all samples increased gradually. Compared to COS, the chitooligosaccharide salt derivatives had stronger scavenging ability. For instance, at the concentration of 1.60 mg/mL, the scavenging effects were as follows: GLCOS—72.75%, FUCOS—91.01%, CMCOS—100%, CFCOS—100%, PCCOS—100%, SPCOS—100%, SYCOS—73.54%, COS—51.85%. In brief, the introduction of phenolic acid can obviously improve the scavenging of hydroxyl radicals, especially the scavenging ability of CMCOS, CFCOS, PCCOS, and SPCOS.

The hydroxyl-radical scavenging activity of COS and phenolic-acid-acylated chitooligosaccharide derivatives is shown in Figure 7b. It can be concluded that the scavenging ability exhibited a concentration-dependent rise in all tested products and reached a maximum value at 1.60 mg/mL. At the same tested concentrations, the order of scavenging effects against hydroxyl radical was as follows: SY-COS > PC-COS > GL-COS > CF-COS > FU-COS > CM-COS > SP-COS > VC > COS. Specifically, the scavenging effects of GL-COS, PC-COS, and SY-COS were nearly 100% in all tested concentrations, which demonstrates the significance of the introduction of phenolic acids into COS.

#### 3.2.4. Reducing Power

Figure 8a shows the reducing power of COS and phenolic acid chitooligosaccharide salt derivatives. Several conclusions can be drawn from the figure. Firstly, the reducing power of phenolic acid chitooligosaccharide salts is obviously higher than that of COS. Secondly, the absorbance of all samples exhibited a concentration-dependent increase. Moreover, at the concentration of 1.60 mg/mL, the absorbances of GLCOS, SPCOS, CFCOS, and PCCOS were higher than that of VC.

Reducing power of COS and phenolic-acid-acylated chitooligosaccharide derivatives is shown in Figure 8b. Similarly to salt derivatives, the phenolic-acid-acylated chitooligosaccharide derivatives had stronger reducing power than COS at all tested concentrations, but the rate of increase was lower than that of the salt derivatives. The experimental results show that the introduction of phenolic acids can obviously enhance the reducing power of COS.

As mentioned above, we can see that both the phenolic acid chitooligosaccharide salt derivatives and the phenolic-acid-acylated chitooligosaccharide derivatives have enhanced antioxidant activities compared to COS. Possible explanations could be summarized as follows. Above all, phenolic hydroxyl groups on phenolic acids are the main groups that exert antioxidant activity. As direct radical scavengers, the hydroxyl groups on phenolic acids can react with free radicals to form a more stable semi-quinone. In addition, the structure of the electron delocalization aromatic nucleus is conductive to the reaction, because it can make the free radical products more stable through the resonance effect of an aromatic nucleus [38]. Furthermore, it can be seen in the results that the antioxidant activity of phenolic acid chitooligosaccharide salt derivatives is stronger than that of phenolic-acid-acylated chitooligosaccharide derivatives. The possible reasons are as follows. When dissolved in the water solution, the salt derivatives can dissociate into ionic forms, which means the higher positive charge density. Additionally, the positive charge is beneficial to scavenging radicals, because it can attract electrons of free radicals and inhibit their chain reaction [37]. Therefore, the antioxidant ability of salt derivatives is stronger than that of acylated derivatives. In addition, among these products, GLCOS has the strongest antioxidant capacity, which was demonstrated in the antioxidant capacity being related to the number of hydroxyl groups: the more hydroxyl groups, the higher the activity. Comprehensively, the experimental results prove that the synthetic strategy of this work preliminarily achieved the expected purpose.

### 3.3. Cytotoxicity Assay

The cytotoxicity of COS and phenolic acid chitooligosaccharide derivatives was detected by L929 cells in vitro, and the results are shown in Figure 9. As shown in the Figure, the cell viability of all samples exceeded 85%, except for 500 and 1000 μg/mL of GLCOS. Moreover, with the increase in concentration, the cell viability of FUCOS, CMCOS, CFCOS, and SPCOS gradually increased. At the concentration of 1000 μg/mL, the cell viability of FUCOS, CMCOS, CFCOS, and SPCOS was more than 100%. This demonstrated the growth inhibition of samples on the cells is weak enough that it could be ignored [39,40]. Comprehensively, the results confirmed the good biocompatibility of the phenolic acid COS derivatives.

The cytotoxicity of COS and phenolic-acid-acylated COS derivatives is shown in Figure 10. Similarly to the COS salt derivatives, the cell viability of all acylated COS derivatives was more than 80% at the tested concentrations. Furthermore, the cell viability of GL-COS, FU-COS, CM-COS, CF-COS, PC-COS, SP-COS, and SY-COS at several tested concentrations was more than 100%. Therefore, the experimental results demonstrate the good biocompatibility of phenolic-acid-acylated COS derivatives. Additionally, Figure 11 and Figure 12 show the L929 cells after adding the two series of derivatives for 24 h (the sample concentration in the cell growth pictures was 1000 μg/mL).

## 4. Conclusions

In this work, two series of phenolic acid chitooligosaccharide derivatives were successfully synthesized through two different synthetic methods. The phenolic acid chitooligosaccharide salt derivatives were innovatively synthesized via ion-exchange method. Additionally, the phenolic-acid-acylated chitooligosaccharide derivatives were synthesized by the EDC/NHS catalytic system. The structures of the derivatives were confirmed by FT-IR and ^1^H NMR spectra. The in vitro antioxidant activity experiments, which included DPPH-radical scavenging activity, hydroxyl-radical scavenging activity, superoxide-radical scavenging activity, and reducing power, were carried out to evaluate the antioxidant capacity of COS and these derivatives. The experiment results indicated that by introducing different phenolic acids to COS, the antioxidant capacity of COS derivatives was improved significantly. The phenolic hydroxyl group processed in phenolic acids plays a key role in improving the antioxidant capacity of the compounds. In addition, the phenolic acid COS salt derivatives have much stronger antioxidant activity than the phenolic-acid-acylated COS derivatives, which can be illustrated by the denser positive and negative charges they have. Furthermore, the cytotoxicity test on L929 cells showed that the derivatives had low cytotoxicity and good biocompatibility. In addition, the derivatives have better water solubility than phenolic acids. In conclusion, the results of all assays substantiated that the chemical modification of phenolic acids significantly influenced the antioxidant activity of COS. This work provides a possible method for developing further uses of COS and phenolic acids. It also provides a new green method for producing biodegradable antioxidant agents that might be applied in the food and cosmetics industries. However, more studies are needed in the future, such as further improving the degrees of substitution of the acylated products, testing the antioxidant activity of derivatives in vivo, and converting them into products with practical value.

## Data Availability

All data contained in the manuscript are available from the authors.

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
