# Peer review of "Synthesis, Characterization, and the Antioxidant Activity of Phenolic Acid Chitooligosaccharide Derivatives"

_marinedrugs, 2022, doi:10.3390/md20080489_

Round 1

Reviewer 1 Report

The manuscript "Synthesis, characterization, and the antioxidant activity of 2 phenolic acid chitooligosaccharide derivatives", was revised. The work sounds interesting. However, it is not clear the novelty of the work. In the introduction section, authors should improve the information and define the state of the art. It is recommended for authors to read the following literature: 

Vo et al. 2017. The free radical scavenging and anti-inflammatory activities of gallate-chitooligosaccharides in human lung epithelial A549 cells. Process Biochem, 54: 188-194. dx.doi.org/10.1016/j.procbio.2017.01.001

Liaqat & Eltem. 2018. Chitooligosaccharides and their biological activities: A comprehensive review. Carbohydrate Polymers, 184: 243-259. https://doi.org/10.1016/j.carbpol.2017.12.067

Mittal et al. 2022. Chitooligosaccharide Conjugates Prepared Using Several Phenolic Compounds via Ascorbic Acid/H2O2 Free Radical Grafting: Characteristics, Antioxidant, Antidiabetic, and Antimicrobial Activities. Foods 2022, 11, 920. https://doi.org/10.3390/foods11070920

Guo et al. 2018. Enhancing the production of phenolic compounds during barley germination by using chitooligosaccharides to improve the antioxidant capacity of malt. Biotechnol Lett (2018) 40:1335–1341, https://doi.org/10.1007/s10529-018-2582-8

Eom et al. 2012. Synthesis of phenolic acid conjugated chitooligosaccharides and evaluation of their antioxidant activity. Environmental Toxicology and Pharmacology, 34: 519-527. dx.doi.org/10.1016/j.etap.2012.05.004

Li et al. 2020. Highly efficient free radical-scavenging property of phenolic-functionalized chitosan derivatives: Chemical modification and activity assessment. International J of Biological Macromolecules, 164: 4279-4288. https://doi.org/10.1016/j.ijbiomac.2020.08.250

Li et al. 2020. Phenolic-containing chitosan quaternary ammonium derivatives and their significantly enhanced antioxidant and antitumor properties. Carbohydrate Research, 498, 108169. https://doi.org/10.1016/j.carres.2020.108169

The authors suggest that a value of cell viability of >80% is "so weak" and could be ignored? Please explain this. Information about possible repercusions in biological systems needs to be discussed.

The conclusion section needs to be improved.

Author Response

Response to Reviewer 1 Comments

Dear Sir or Madam,

Thank you for your comments concerning our manuscript entitled “Synthesis, Characterization, and the Antioxidant Activity of Phenolic Acid Chitooligosaccharide Derivatives”. Those comments are all valuable and very helpful for revising and improving our paper. We have studied comments carefully and have made corrections which we hope meet with approval. The main corrections in the manuscript and the responds to the comments are as following:

Point 1: English language and style: English language and style are fine/minor spell check required.

Response 1: Thank you for your kind suggestions and according to your suggestions, we carefully checked the English expressions and spellings in the manuscript and made corrections. The revision has been indicated in the revised manuscript.

Point 2: Does the introduction provide sufficient background and include all relevant references? Must be improved.

Response 2: Thank you for your kind suggestions and according to your suggestions, we have added the background information related to our study in the Introduction part (lines 92-126 of the revised manuscript).

Point 3: Are the conclusions supported by the results? Must be improved.

Response 3: Thank you for your kind suggestions and according to your suggestions, we have modified the section of conclusions and the revision has been indicated in the manuscript (lines 504-537 of the revised manuscript).

Point 4: The manuscript "Synthesis, characterization, and the antioxidant activity of 2 phenolic acid chitooligosaccharide derivatives", was revised. The work sounds interesting. However, it is not clear the novelty of the work. In the introduction section, authors should improve the information and define the state of the art. It is recommended for authors to read the following literature.

Response 4: Thank you for your kind suggestions according to your recommendation, we have carefully read the recommended literatures and cited them in the revised manuscript. As for the novelty of the work, in our study, seven new phenolic acid COS salts derivatives were innovatively synthesized by ion exchange method. Besides, the phenolic acid acylated COS derivatives were synthesized by EDC/NHS system. The antioxidant activity of the two series of derivatives was obviously stronger than COS, and it is observed that the antioxidant activity of salt derivatives was higher than that of acylated derivatives. We have revised the manuscript accordingly and hope this explanation will answer your confusion.

Point 5: The authors suggest that a value of cell viability of >80% is "so weak" and could be ignored? Please explain this. Information about possible repercusions in biological systems needs to be discussed.

Response 5: Thank you for your kind suggestions and according to your suggestions, we have made some modifications in this part. After literature exploration, we have revised this part to: the cell growth inhibition was weak enough that could be ignored (lines 474-482 of the revised manuscript).

Point 6: The conclusion section needs to be improved.

Response 6: Thank you for your kind suggestions and according to your suggestions the conclusions have been modified and the revision has been indicated in the manuscript (lines 504-537 of the revised manuscript).

The revised manuscript has been submitted to your journal. We hope that the responses and the revised manuscript adequately address your concerns and that this revised version is now acceptable for publication. Thank you for your time and concerns. If you have any questions, please feel free to contact us.

Yours sincerely,

Zhanyong Guo

Reviewer 2 Report

The Introduction of manuscript can be improved. Similar publication published in 2012

Tae-Kil Eom, Mahinda Senevirathne, Se-Kwon Kim, Synthesis of phenolic acid conjugated chitooligosaccharides and evaluation of their antioxidant activity, Environmental Toxicology and Pharmacology, 2012, 34, 519-527 has not been commented in the Introduction.

Dae-Sung Lee, Ji-Young Woo, Chang-Bum Ahn, Jae-Young Je, Chitosan–hydroxycinnamic acid conjugates: Preparation, antioxidant and antimicrobial activity, Food Chemistry,2014, 148, 97-104 also has not been commented.

The comparison of results in presented manuscript with the results generated in above mentioned publications and discussion  of the progress  during the last 10 years  would be beneficial. 

FTIR spectra in the region 2000cm-1  -  4000cm-1 are not commented.

It would be beneficial to describe in more detail challenges and future perspectives.

Author Response

Response to Reviewer 2 Comments

Dear Sir or Madam,

Thank you for your comments concerning our manuscript entitled “Synthesis, Characterization, and the Antioxidant Activity of Phenolic Acid Chitooligosaccharide Derivatives”. Those comments are all valuable and very helpful for revising and improving our paper. We have studied comments carefully and have made corrections which we hope meet with approval. The main corrections in the manuscript and the responds to the comments are as following:

Point 1: English language and style: Moderate English changes required.

Response 1: Thank you for your kind suggestions and according to your suggestions we carefully checked the English expressions and spellings in the manuscript and made corrections. The revision has been indicated in the revised manuscript.

Point 2: Does the introduction provide sufficient background and include all relevant references? Can be improved.

Response 2: Thank you for your kind suggestions and according to your suggestions we have added the background information related to our study in the section of Introduction (lines 92-126 of the revised manuscript).

Point 3: The Introduction of manuscript can be improved.

Response 3: Thank you for your kind suggestions and according to your suggestions we have added relevant research background content in the Introduction part.

Point 4: FTIR spectra in the region 2000cm-1 - 4000cm-1 are not commented.

Response 4: Thank you for your kind suggestions and according to your suggestions we have added the analysis of FTIR spectra of the region 2000cm-1 - 4000cm-1 (lines 299-301 of the revised manuscript).

Point 5: It would be beneficial to describe in more detail challenges and future perspectives.

Response 5: Thank you for your kind suggestions and according to your suggestions we have described the challenges and future perspectives in more detail in the manuscript (lines 528-531 of the revised manuscript).

The revised manuscript has been submitted to your journal. We hope that the responses and the revised manuscript adequately address your concerns and that this revised version is now acceptable for publication. Thank you for your time and concerns. If you have any questions, please feel free to contact us.

Yours sincerely,

Zhanyong Guo